# Functionalization of and through Melanin: Strategies and Bio-Applications

**DOI:** 10.3390/ijms24119689

**Published:** 2023-06-02

**Authors:** Alexandra Mavridi-Printezi, Arianna Menichetti, Dario Mordini, Marco Montalti

**Affiliations:** Department of Chemistry “Giacomo Ciamician”, University of Bologna, Via Selmi 2, 40126 Bologna, Italy; alexandra.mavridi2@unibo.it (A.M.-P.); arianna.menichetti2@unibo.it (A.M.); dario.mordini2@unibo.it (D.M.)

**Keywords:** melanin, polydopamine, nanoparticles, nanomedicine, photothermal therapy (PTT), photosensitizer (PS), photothermal agent (PTA), chemotherapy (CT), gene therapy (GT), cancer

## Abstract

A unique feature of nanoparticles for bio-application is the ease of achieving multi-functionality through covalent and non-covalent functionalization. In this way, multiple therapeutic actions, including chemical, photothermal and photodynamic activity, can be combined with different bio-imaging modalities, such as magnetic resonance, photoacoustic, and fluorescence imaging, in a theragnostic approach. In this context, melanin-related nanomaterials possess unique features since they are intrinsically biocompatible and, due to their optical and electronic properties, are themselves very efficient photothermal agents, efficient antioxidants, and photoacoustic contrast agents. Moreover, these materials present a unique versatility of functionalization, which makes them ideal for the design of multifunctional platforms for nanomedicine integrating new functions such as drug delivery and controlled release, gene therapy, or contrast ability in magnetic resonance and fluorescence imaging. In this review, the most relevant and recent examples of melanin-based multi-functionalized nanosystems are discussed, highlighting the different methods of functionalization and, in particular, distinguishing pre-functionalization and post-functionalization. In the meantime, the properties of melanin coatings employable for the functionalization of a variety of material substrates are also briefly introduced, especially in order to explain the origin of the versatility of melanin functionalization. In the final part, the most relevant critical issues related to melanin functionalization that may arise during the design of multifunctional melanin-like nanoplatforms for nanomedicine and bio-application are listed and discussed.

## 1. Introduction

In recent decades, the use of nanomaterials has revolutionized many fields related to human preservation and well-being, spanning from medicine and therapy to cosmetics and food [1,2]. One of the unique features of nanomaterials is that their intrinsic physical, chemical, and biological properties can be easily tuned and expanded by surface functionalization. Exploiting surface functionalization allows for multifunctional nano-systems to achieve easily: (i) modifying their original surface chemical termination, (ii) adding new functional molecules on their surface, or (iii) covering their surface with an additional coating [3,4]. Surface modification is hence a strategy that can endow versatility to a simple nanoplatform and control important features for its bio-application, improving drawbacks such as non-optimal selectivity of interaction with the target tissue or biomolecule or non-complete biocompatibility [5]. After being surface-modified, the new nanoplatform can show multiple actions, combining therapy with diagnosis in a theragnostic approach [6,7]. Additionally, functional coatings can act as a shielding matrix for drugs or bioactive agents, protecting them from degradation and controlling their bioavailability and release [8]. Melanin, both in its natural and synthetic forms, is an ideal material for the production of multifunctional nanostructures since it possesses unique optical and electronic properties, and it can be functionalized by simple, effective, and easily tunable approaches.

Melanin is a family of bio-pigments with diverse origins and structures that play various fundamental roles in nature, including photoprotection from the harmful outcomes of irradiation and coloration [9]. Being essential features for bio-application, melanin extracted from natural sources possesses inherent biocompatibility and biodegradability and can be extracted already as nanoparticles (NPs) [10,11]. Melanin, in fact, is one of the rare materials found in living organisms in the form of NPs, e.g., as melanosomes or in the ink of cuttlefish and other mollusks. In the same manner, synthetic melanin-like NPs (MNPs) combine the unique prominent features of the natural pigment with simple and mild preparation procedures appearing as promising candidates for the development of new nanoplatform for diverse biomedical applications [12,13]. Indeed, in recent years, extraordinary research developments based on naturally extracted melanin NPs and MNPs have been achieved, indicating the significance of this family of materials [14,15,16,17,18,19]. Moreover, synthetic melanin can be easily produced in the form of film or coating of existing NPs made by other materials (e.g., metals or metal oxides). Therefore, synthetic melanin is an ideal functionalization candidate since it can create functional mussel-inspired coatings almost on any surface in a buffered solution without the use of expensive or harsh chemicals [20].

Functionalization is a powerful tool for the application of NPs in medicine, especially when multi-functionality is needed, such as in theragnostic applications, and it can be exploited to increase the multiplicity of the therapeutic system and the modalities of bioimaging [21]. With proper functionalization, NPs can act as nanocarriers for targeted therapy showing high selectivity, reduced side effects on normal tissues, enhanced cellular penetration and accumulation, high loading capacity, and controlled release of actives and drugs [22,23]. This strategy can also be exploited to overcome the drug-resistance of some organisms due to the excessive use of drugs (e.g., antibiotics) in the modern age [24].

The functionalization of NPs has allowed chemical therapy to be easily combined with other modes of treatment. This multitherapeutic approach is particularly important in the case of cancer, a disease that has been recognized as a major threat to human health. Indeed, multifunctional nanosized therapeutic platforms have shown great potential in tumor ablation and cancer inhibition, indicating potential applicability to cancer therapy [25]. With the capacity to reduce the cytotoxicity of conventional chemotherapeutic drugs, multi-functional NPs have presented promising results in novel minimal-invasive therapies such as photothermal therapy (PTT), photodynamic therapy (PDT), and immunotherapy [26,27]. PTT represents a non-invasive therapeutic modality with a high temporal-spatial resolution which is based on the use of photothermal transduction agents (PTA) able to generate localized heat after being irradiated [28]. In the case of melanin NPs, because of their optical and electronic properties, they can be exploited as a very effective PTA while other functions (e.g., targeted drug delivery) can be integrated by functionalization. Melanin is a very important PTA since it presents a large absorption coefficient in the near-infrared region (NIR), and hence, an enhanced tissue penetration of irradiation can be achieved. Additionally, in PTT, the damage to healthy tissue can be minimized by controlling the light-triggered region [29]. In the case of PDT, another constantly developing therapeutic strategy are specific photo-excited photosensitizers (PS), which interact with oxygen-generating cytotoxic reactive oxygen species (ROS), which can kill cancer cells [30]. In this case, melanin is not suitable to act as a PS, but NPs can be easily converted into PS by functionalization with the proper photo-responsive units. In this review, we will show how these functionalized NPs can easily combine with PDT with PTT and can be further functionalized to perform simultaneous chemical therapy and/or target recognition.

Undoubtably, combining a therapeutic action with imaging possibility is highly advantageous for monitoring the efficiency of drug delivery, visualizing off-target effects and possible toxicity while observing the therapeutic outcomes of the method. In view of this, numerous nanotheragnostic agents have been suggested as liposomes, polymeric NPs, or coated-metal NPs for a variety of imaging-guided techniques such as Magnetic Resonance Imaging (MRI), Computed Tomography (CT) and photoacoustic imaging (PAI) [31,32,33]. Compared to other nanomaterials, biomimetic nanomaterials, such as melanin-based ones, are highly advantageous since they present intrinsic biocompatibility and biodegradability, decreasing side effects and exhibiting, in some cases, functions that may mimic biological systems [34,35]. In this review, we will show how exploiting functionalization, melanin-based nano-therapeutic agents can also be modified to behave as efficient contrast agents for bioimaging, e.g., by suitable loading with metal ions.

The versatility of the functionalization of melanin arises from the presence of abundant chemically active functional groups that can be covalently or non-covalently modified through simple reactions. For this reason, creating a shell of melanin on pre-existing NPs of other materials (e.g., metal or metal oxide) is a valid strategy to functionalize non-melanin-like NPs. This approach exploits the tendency of melanin-like materials, e.g., polydopamine (PDA), to adhere to almost any surface, forming a controlled layer in very mild and environmentally friendly conditions. In the second step, the shell on the outer layer of NPs can be easily functionalized by various desired molecules [36,37]. Another possible strategy for functionalizing MNPs is the control of their assembly. In this approach, melanin can be bound with other bioactive components giving rise to new building blocks that can co-assemble, eventually forming new MNPs able to respond, e.g., with disaggregation, to specific biological environments [38,39].

In this review, an overview of all the surface-functionalization possibilities of melanin or through melanin, hence using synthetic melanin as a coating, will be given to summarize all the feasibilities that this family of materials can offer, discussing the most recent and relevant applications related to biology and medicine. This work is mainly focused on NPs, and different methods of pre- and post-functionalization will be discussed to clarify the structural and physicochemical differences between the various melanin-like materials and distinguish between melanin-modified NPs and other NPs coated with PDA or differentiate melanin co-polymers. A clear division between post- and pre-synthetic (co-polymers) strategies will also be maintained in the discussion of some of the most recent and significant approaches developed in recent years.

### 1.1. Melanin and Melanin-like Materials

#### 1.1.1. Natural Melanin

Natural melanin constitutes a big family of heterogenous biopigments diverse in origin and structure. Even though the existence and importance of melanin have been known for decades, and despite it being a biomaterial widely studied, up to now, the complete structure and composition of the subunits of the pigment are still missing. Notwithstanding, melanins are highly divergent, it is widely accepted that they all share the same physicochemical and optical properties, and they derive from the enzymatic oxidation and polymerization of tyrosine precursor in animals or other phenolic derivatives in microorganisms [14,40]. The fascinating properties of melanin include (i) broadband monotonic UV–Vis (ultraviolet-visible) absorption that also extends to the NIR responsible for the photoprotective role of the pigment, (ii) antioxidant (AOX) and free radical scavenging activity due to the presence of various functional groups such as catechol, amine, and imine able to offer defense against a wide range of solar-induced or disease-related ROS, (iii) paramagnetic properties presenting persistent electron spin resonance (ESR), (iv) high drug loading capacity thanks to the presence of π-conjugated structures, and (v) metal-chelation ability [9,11,41,42]. Even if photoprotection is one of the most popular and important properties of the pigment in animals, in many cases, melanin has been correlated to phototoxicity, and even cancer, indicating the need to unravel the complete structure and polymerization pathway of the pigment for a better understanding of its properties [13]. In animals, therefore, also humans, the only types of pigment produced are dark-brown eumelanin, red-yellow pheomelanin, and neuromelanin. In humans, during melanogenesis, the process of melanin synthesis starts with tyrosine; the pigment is produced in the form of specific spherical organelles known as melanosomes [43]. During tyrosine-derived melanin synthesis, a step difference is involved in the production of eumelanin and pheomelanin, resulting in two pigments with not only different structures but also properties. In more detail, even if the supramolecular structures are still unknown, eumelanin is considered to be a heterogeneous polymer consisting of 5,6-dihydroxyindole (DHI) and 5,6-dihydroxyindole-2-carboxylic acid (DHICA) in various ratios, while, since during pheomelanin synthesis cysteine is incorporated at positions 2 or 5 of the benzene ring, pheomelanin is composed by benzothiazine and benzothiazole units [44]. Something that should not be omitted is that eumelanin is related to photoprotection as it absorbs the UV–Vis radiation dissipating the energy harmlessly, while pheomelanin is considered photoreactive and phototoxic. Regarding microbes, except for the two aforementioned types, they also possess two additional types of pigment known as pyomelanin and allomelanin [10]. Allomelanin is derived from 1,8-dihydroxynaphthalene (1,8-DHN), and it is a nitrogen and sulfur-free melanin, while pyomelanin arises from the catabolism of tyrosine through the oxidation and self-polymerization of homogentisic acid (HGA) [15].

Natural melanin can be extracted from various sources, including human hair, bird feathers, black sesame seeds, marine organisms, and microbes [10,14,45,46]. Since melanin, in some cases, is bound to other cellular components, it can be challenging to isolate it. For this reason, extraction methods may involve the use of harsh chemicals that can alter the pigment in an uncontrolled way, affecting its physicochemical properties. Moreover, inconsistency is observed between the natural samples as the biological synthesis of the pigment is affected by various factors such as the type of the organism, the living environment, and seasonal changes [47,48]. As an example, the synthesis of fungal melanin can be affected by inhibitors, carbon sources, and nitrogen sources; however, specific genes can be used to regulate and control the synthesis of melanin by the microbes in order to have a better-controlled formation of the pigment. Different from other extraction sources, sepiamelanin can be extracted from the ink sac of cuttlefish simply by centrifugation of the black ink obtained by opening the sac of the fish, while washing is the only treatment step required to retrieve spherical NPs from the fish [49]. For this reason, sepiamelanin is the most widely used naturally extracted melanin and has been recognized to be a standard natural eumelanin-like material exploited in a variety of bioapplications [11,50].

#### 1.1.2. Synthetic Melanin-like Materials

Even if sepiamelanin is widely exploited due to its inherent biocompatibility and ease of extraction, in this case, the kind of animal, the geographic region, and the season may affect the extracted pigment mitigating the reproducibility and the consistency between samples. For this reason, the synthesis of melanin-like materials represented a real breakthrough because they could mimic the unique physicochemical properties of the natural pigment, being highly biocompatible and offering the possibility to control their formation and synthetic condition [51]. Additionally, since the complete structure and the synthetic passages of the natural pigment are still obscure, synthetic melanin analogs have been exploited as a model for the understanding of natural melanin formation and the correlation between its structure and its properties [40,52]. Among their properties, the high photothermal conversion efficacy that melanin-like materials possess is one of the most exploited ones since they are ideal biocompatible PTAs showing large absorption coefficients in the NIR and ultrafast energy dissipation through non-radiative pathway [53,54]. Up to now, melanin-like materials can emerge in various forms that include NPs [55], surface adherent films [56], and high molecular weight water-soluble aggregates and polymers [57], while three main approaches have been exploited so far for their fabrication which are enzymatic oxidation, solution oxidation by atmospheric oxygen or chemical oxidants, and electro-polymerization [58,59,60,61,62]. To begin with, polydopamine (PDA) is the most exploited melanin-like material due to its simplicity of preparation, easy scalability, and tunability of the final product. In more detail, PDA is considered to be the synthetic analog of natural eumelanin and can be synthetized directly in the form of spherical, monodispersed NPs by the oxidation and self-polymerization of dopamine (DA) in alkaline and aerobic condition in ethanol/water solvent [63]. Ammonia (NH_3_) and Sodium hydroxide (NaOH) can be utilized for the pH adjustment and the size of the NPs can be easily tuned by adjusting the DA/base ratio in the case of NH_3_, while temperature control is also needed in the case of NaOH (Figure 1a) [52]. Upon modulation of both synthetic protocols, the desired size can be chosen from 10 to 400 nm resulting in an NP population with narrow size distribution every time and thus high monodispersity reaching polydispersity index (PDI) values of lower than 0.05 [40,64,65,66]. Indeed, synthetic MNPs have shown a narrower size distribution compared to naturally extracted sepia melanin NPs in which various populations of size from 100 to 300 nm are present in the same environmental sample [67]. What is particularly interesting is that PDA NPs can be incorporated by human keratinocytes in the same way as the organism’s natural melanosomes forming the so-called perinuclear carp to protect the genetic material from photodamage [63]. Except for solid spherical PDA NPs, mesoporous PDA NPs can be synthesized by exploiting the use of a template which, in a second step, is solubilized and removed. Pluronic F127 is the most widely utilized template for the synthesis of mesoporous PDA NPs, as it can be easily removed using a mixture of ethanol/acetone [68,69]. These kinds of MNPs are particularly advantageous because they possess abundant porous cavities which can effectively load many drugs and actives, serving as ideal biocompatible nano-carriers. Except for the two aforementioned types, hollow PDA NPs can also be synthetized using SiO_2_ NPs as a template. For their synthesis, a PDA coating is formed on SiO_2_ NPs by the polymerization of DA in alkaline buffer solution and then the NPs are treated hydrothermally within a Teflon-lined autoclave at 200 °C to allow the formation of hollow/yolk-shell structures [70].

DA is the most used precursor for the formation of PDA MNPs; however, 3,4-Dihydroxy-L-phenylalanine (L-DOPA), being also a precursor in the human melanogenesis pathway, can be utilized for the synthesis of MNPs. In this case, oxidation by oxygen in NH_3_ solution does not lead to the formation of NPs but to the formation of a water-soluble polymer that can be precipitated with acetonitrile (ACN) and has been reported to have a graphite-like structure [57]. For the formation of MNPs starting from L-DOPA, stronger oxidants such as potassium permanganate (KMnO_4_) are essential. Another alternative is the exploitation of the enzyme-catalyzing natural melanin synthesis, where tyrosinase can catalyze the formation of L-DOPA NPs in buffered solution, also giving the possibility to incorporate cysteine for the formation of artificial pheomelanin NPs [48].

Apart from mammalian synthetic melanin-like analogs, fungal analogs have been synthesized and reported to possess superior AOX activity after being incorporated into human keratinocytes [71]. For the synthesis of fungal melanin-mimicking allomelanin NPs, the nitrogen-free precursor 1,8-dihydroxynaphthalene (1,8-DHN) is polymerized at ambient temperature in the presence of either sodium periodate (NaIO_4_) or KMnO_4_. In this case, the morphology of the NPs can be tuned by tuning the NaIO_4_ to 1,8-DHN monomer ratio leading from spherical to walnut-like shapes by increasing the amount of the oxidant. Moreover, starting from the self-polymerization of different DHN dimers, namely 4-4′, 2-4′, and 2-2′ DHN dimers, the morphology and the molecular structure of the self-assembled structure is significantly affected as the pre-organization of the two substituted naphthalene rings determines the final type of allomelanin formation [72]. Finally, 24h aging of standard allomelanin NPs produced by NaIO_4_ can lead to the occurrence of hollow NPs after being dispersed for two days in methanol, while lacey NPs can appear after 6 days in the same solvent [73]. No matter the type of synthetic MNPs, so far, a negative Z-Potential, thus a negative surface charge at neutral pH, has been indicated for all of them, explaining their high colloidal stability [52,63,71].

Among the melanin-like materials, PDA films are also included and have been widely exploited in a wide range of fields, from medicine to catalysis [74,75,76]. Adherent mussel-inspired PDA films can be formed onto almost any kind of surface by simple immersion of the desired substrate in an alkaline buffered DA solution (pH = 8.5) (Figure 1b) [74]. Two critical parameters affect the deposition kinetics of the PDA film on a surface which are: (i) the type of buffer and (ii) the type of oxidant [60]. However, the final chemical structure of the film is independent of the substrate because even at the early stages of deposition, a different PDA chemical composition is observed when comparing various substrates. At the later and final stages of layer formation, a unified structure is reported [77]. In the same manner, PDA can be utilized as an external coating of various NPs as metallic or polymeric NPs and can confer to them enhanced biocompatibility and photothermal conversion properties (Figure 1c) [78,79,80].

### 1.2. Melanin Surface Chemistry Governs the Functionalization Capability

Even if a considerable amount of research has been conducted by the scientific community, up to now, the complete structure of melanin and the chemical composition of the subunits are still not completely explored [57]. Even though this hinders the complete understanding and full exploitation of the potential of this material, it is well known that melanin possesses a variety of surface active functional groups that can act as target sites for functionalization [12]. In particular, the presence of multiple catechol groups, quinones, and amino functionalities gives rise to various kinds of interaction, such as hydrogen bonding, π–π stacking, coordination, and covalent binding (Figure 2) [42]. To begin with, the Michael addition can be exploited for the covalent functionalization of melanin with nucleophile thiols, while the Schiff base reaction can be performed for amino-functionalization [81]. As mentioned before, melanin and, in a similar manner, also PDA result from the oxidative polymerization and ring-closing of the precursor. The quinone groups arising from the catechol, such as DA, oxidation are the most valuable moieties that can react with a variety of thiols and amines present in biomaterials as proteins and oligonucleotides, but also synthetic molecules giving rise to novel PDA-conjugates (Figure 2) [82]. More specifically, o-quinones, exhibiting carbonyl and Michael acceptor properties, can undergo 1,2-and/or 1,4-addition [55]. Moreover, the π-conjugated structures on melanin’s surface can bind with various actives having aromatic structures via π–π stacking, reversible hydrogen bonds, and electrostatic interactions [83]. Carbodiimide reactions cannot be excluded because the amino group-rich PDA surface can react with molecules possessing a carboxylic group, while aryl–aryl covalent bonding can also occur [18]. Except for the huge scientific effort that has been performed for the identification of the surface chemistry of both melanin and PDA, their similar surface chemistry is already denoted by human cells [41,55,77,84]. Specifically, the similar surface functionalities of melanin and PDA NPs are revealed by living cells as the NPs are able to be incorporated by human keratinocytes as “self” indicating that the surface chemistry of the melanin analogue governs the cellular uptake as the cells “confuse” them with their own melanosomes [63].

### 1.3. Pre- and Post-Functionalization of Melanin

The ease of functionalization of melanin and melanin-like materials led to the development of numerous novel nanoagents with extraordinary properties [85]. Before going into detail on the benefits that functionalization can endow melanin with, two different approaches should be introduced. Importantly, even if these two strategies involve the same reagents, they can lead to completely different surface properties. In particular, these two approaches are known as “secondary” or “post-” functionalization and “one-pot” or “pre-” functionalization [82,84,85]. Post-functionalization refers to the modification and processing of the surface of synthesized materials, or of already formed NPs, by physical and chemical means in order to provide new functions to it (Figure 2) [86]. Naturally occurring melanin, such as sepia melanin, can be retrieved from living organisms and functionalized in the laboratory by chemical means [50]. In the case of melanin-like materials, this modification is conducted after the formation and purification of the synthetic pigment, which can emerge either in the form of NPs or as a film [85,87]. However, the term post-modification may also refer to the use of PDA as a surface modification strategy of other materials, known as PDA-based surface modification, where synthetic eumelanin acts as a functional coating [88]. Regarding pre-modification (Figure 3), this approach involves the co-polymerization of the precursor with another molecule that will partially modify the resulting material maintaining its fundamental characteristics but providing new or enhanced functions. While the homopolymer of PDA is usually used as a functional coating, co-polymers of DA with catechol or amine-containing molecules can exhibit extraordinary properties [89]. It should be clarified that natural melanin, because it is synthesized by living organisms and extracted from them, cannot undergo pre-functionalization, but only post-treatment can occur.

In this review, an overview of all the surface modification possibilities of or through synthetic and natural melanin is presented. Many nice and extensive reviews have already been published focusing either on a particular application of melanin as cancer treatment, either in the use of melanin-coatings or NPs, without distinguishing whether they are referring to melanin NPs or PDA-coated NPs or melanin co-polymers [12,19,42,85]. Indeed, in most cases, the nanocarriers are simply mentioned as melanin-like nanoparticles (MNPs) without distinguishing between functionalized PDA NPs and PDA-coated other types of NPs. In this review, we want to provide an overview of all the possibilities that can emerge using synthetic and natural melanin, classifying the different applications based on whether melanin acts as a coating or as a cover material and whether it is pre- or post-functionalized. Indeed, the chapter separation is based on the surface modification strategy instead of the application/purpose of the developed melanin-based nano-agents (Figure 2 and Figure 3). Particular attention is given to the post-modification of natural melanin and MNPs. The most recent examples of important surface modification strategies of the pigment will be described indicating the improved properties that each particular modification can endow on the developed agent. Approaches able to improve target specificity, increase biocompatibility and provide a synergistic effect will be the center of attention. Regarding the pre-modification of synthetic melanin, various co-polymers of melanin-like materials will also be presented since they can exhibit superior properties and a more specific action than melanin-homopolymers. In addition, a short overview of the utilization of PDA as a functional coating of other kinds of materials is also provided, as PDA presents a valuable surface-modification strategy that is widely exploited by the scientific community for the coating of NPs or substrates.

## 2. Functionalization

Surface modification of materials has attracted much attention from the scientific community as it can have bilateral advantages for both the nano-agent and the organism, in cases of bio-application. In particular, surface coatings can protect the internal material from alterations while also providing improved biocompatibility [90]. Additionally, they can offer a secondary reaction platform that can enhance in various ways the properties of a nano-agent [91]. Easy functionalization approaches are always at the center of attention due to their versatility and facile realization without the use of harsh chemical conditions. In view of this, melanin and melanin-like materials are ideal candidates due to their intrinsic biocompatibility, biodegradability, and multifunctionality [42]. As mentioned before, natural and synthetic melanin possess various potential binding or biosorption sides able to bind active molecules that have been exploited for the development of numerous surface-modified multifunctional nano-agents [92]. As a functional coating, PDA can form a mussel-inspired adherent film almost on any surface by simply dip-coating objects in a DA solution at alkaline pH. Due to the presence of numerous functional groups, PDA can also act as a transition layer where secondary reactions take place for the further modification of films [74]. In a similar manner, PDA-based coating of NPs is among the most studied surface-modification approaches especially for bio-application, as it can enhance the biocompatibility and cellular uptake ability of potentially toxic NPs endowing also photothermal properties to the nano-agent [12,93]. Due to the high versatility of PDA, melanin-like co-polymers can also present different surface properties without needing additional post-modification. In this section, an overview of the main post and pre-functionalization approached of melanin and melanin-like materials will be given, while also the utilization of synthetic melanin as a functional coating of other materials will be briefly described.

### 2.1. Post-Functionalization of Melanin NPs and MNPs

Post-functionalization is a valuable tool for the addition of new properties to already formed NPs or for the enhancement of already existing ones. Since the modification is conducted after the synthesis and/or purification of already formed NPs, more than one functionality can be added step-wisely or simultaneously on their surface, while an easy comparative study can be made between pure and functionalized NPs, without the need for new batch synthesis. Moreover, for the same reason, post-functionalization does not affect the size distribution and surface morphology of already synthesized NPs [67]. On the contrary, an increase in the hydrodynamic diameter of post-modified NPs is usually observed by the addition of various functionalities such as PEG, metals, and peptides [94,95,96,97]. The main change upon functionalization is the surface charge and, therefore, the Z-Potential of the NPs. Highlighting the importance of post-functionalization, the most widely exploited strategies of after-synthetic modification of natural and synthetic MNPs are summarized in view of their bio-application. This section is focused on melanin NPs and not melanin-coated NPs (see Section 2.3) and the subdivision is based on the surface modification approach rather than on the application. The most recent examples will be discussed, while comparison will be made, when possible, between similar synthetic protocols indicating possible alternations and differences.

#### 2.1.1. PEGylation for Increased Circulation Time and Improved Target Specificity

Up to now, one of the main challenges of nanocarrier-assisted drug delivery is a prolonged circulation time and good in vivo stability because many vesicles undergo fast immune clearance and degradation [98]. Moreover, satisfactory in vivo stability should be combined with high biocompatibility to avoid organ toxicity upon accumulation of the nanocarrier in them [99]. These issues can be tackled by specific surface modification of nanocarriers that may optimize their blood circulation time and increase their target specificity. Especially, hydrophilic macromolecules and polymers have been at the center of attention as they can form a hydrophilic steric barrier on the surface of particles. Even if a high number of polymers have been tested for the delivery of actives, polyethylene glycol (PEG) is one of the most widely used. The reasons why PEG is so popular are numerous and among its advantages are that (i) it is water-soluble and biocompatible, (ii) it can prolong the circulation time of a nanocarrier by increasing its hydrophilicity and reducing the rate of glomerular filtration, therefore the serum half-life of the active ingredient increases, and so does the therapeutic action, (iii) it improves the pharmacokinetics by protecting the vesicle or the active from reticuloendothelial cells and proteolytic enzymes, and (iv) it enhances the efficiency of the delivery of a drug or active to target cells and tissues [100]. Regarding cancer, PEG-modified nanomaterials exhibited higher accumulation in solid tumors due to their improved permeability and enhanced retention ability [101]. However, a careful choice of reactive PEG should be made because, in some cases, hydrophilic nonionic PEG coatings can have an adverse effect on the NPs and hinder the interaction of the NPs with the cell membrane leading to poor cellular uptake [102]. The surface-modification technique of covalently binding PEG to molecules or NPs is known as “PEGylation”. Knowing that natural and synthetic melanin NPs are promising candidates as nanocarriers for drug delivery, in this section, examples of melanin NPs PEGylation are presented since it is one of the most exploited post-functionalization strategies. The reason why this functionalization strategy is so popular is that it can prolong the blood circulation half-life of unmodified melanin NPs from 4 h to 15.7 h upon modification [103,104]. It should be highlighted that PEGylation is usually the first post-modification strategy applied and, in most of the cases, it is combined with additional surface functionalizations for a synergistic action [101,105,106].

Even if several reactive PEG have been suggested [105], thiol- (SH-PEG) or amine-terminated PEG (NH_2_-PEG) are the most exploited ones for the functionalization of melanin NPs, and therefore Michael addition and Schiff bases are the reactions of choice for the covalent grafting of PEG moieties on the surface of the NPs. This is due to the biosafety of the aforementioned reactive PEG groups and the simplicity of the functionalization reaction of melanin in the absence of harsh conditions. Usually, PEGylated MNPs are obtained simply by mixing NPs with PEG moieties in a buffered solution. In fact, SH-PEG reacts with the catechol, or with the unsaturated indole rings resulting from the oxidation and ring-closing of the melanin precursor, via Michael addition. NH_2_-PEG can also react with melanin NPs through a Schiff base by the condensation reaction between the amine terminus group and the quinone [84]. Except for monofunctional PEG, bi-functional PEG with two reactive ends (either hetero-bifunctional or homo-bifunctional) is highly advantageous since they can tune the hydrophilicity of the functional coating, e.g., with the introduction of charges, as only one of the two sides (-NH_2_ or -SH) will react with the PDA or natural melanin [107]. In order to understand how different PEG functionalities can influence the skin interaction of MNPs by affecting their surface properties, Sunoqrot et al. prepared a set of PEGylated PDA NPs and compared their skin accumulation capacity (Figure 4) [108].

Four different types of PEG-modified PDA NPs were prepared by mixing different functional precursors with purified PDA NPs in a basic NaOH solution. Simple neutral mPEG-PDA was obtained, exploiting the coupling to the thiolate group using PEG–SH. Anionic NPs resulted from functionalization with HS–PEG–COOH, while cholesterol-modified hydrophobic NPs were produced with HS–PEG–Chol (cholesterol). Cationic surface termination was achieved using NH_2_–PEG–NH_2_ that reacted with PDA through the amino group. Strat-M synthetic membranes were used as a skin model for transdermal permeation studies, and the in vitro experiments showed that anionic PEG-PDA NPs exhibited the highest degree of accumulation followed by the neutral and then the positively charged PEG-PDA NPs, while the accumulation efficacy of hydrophobic NPs was minor. It should be highlighted that PDA NPs without modification already possess a negatively charged surface (see Section 1.1.2), and therefore their accumulation capacity is similar to carboxyl-terminated PEG-PDA NPs. However, a considerable difference was observed in ex vivo studies on human skin. In this case, unmodified PDA NPs tended to aggregate, indicating poor steric stability in the presence of proteinaceous skin secretions, and they could permeate the stratum corneum and stay deposited there. On the contrary, all the PEGylated PDA NPs showed good colloidal stability, and particularly the anionic ones were able to pass to the epidermis and then the dermis demonstrating pronounced skin retention.

Although, as mentioned above, PEGylation of PDA is widely exploited for bio-application, the actual control of the functionalization process remains a challenge. To overcome this obstacle, Zmerli et al. proposed the utilization of the heterobifunctional SH-PEG-COOH, which also exhibited the best performance in the aforementioned study, for enhanced cellular uptake and controlled surface functionalization of PDA NPs [97]. The PEGylation was carried out in Tris buffer solution at pH = 8.5 in a ratio of 1:10 for PDA NPs to PEG. By using a Quartz crystal microbalance with dissipation monitoring (QCM-D) in combination with atomic force microscopy (AFM), the PEGylation kinetics were evaluated, finding that the ideal reaction conditions are when Tris is used as a solvent in the presence of salt (NaCl) instead of ethanol/water mixture and that the PEG polymer with the ideal molecular weight was 2000 Da compared to 3500 or 5000 Da. As already mentioned, anchoring of PEG to the NPs was achieved through the thiol group, while the carboxylic group can be exploited to bind new ligands.

PEGylation is an essential post-functionalization strategy of melanin NPs for nanotherapeutics where a prolonged circulation time of the nanoplatform is needed, and it has found applications in cancer treatment and diagnosis, one of the most important challenges for humanity [85,88]. For this application, PEGylation is often combined with additional, secondary modifications for synergistic therapy or theragnostic. For example, PEGylation is combined in many cases with metal [109], gene [110], and peptide [111,112] functionalization, discussed in the next sections, for enhanced multimodal performance. Moreover, drugs can be loaded to MNPs after the PEGylation through π−π stacking and/or hydrogen bonding [113]. For example, PDA NPs functionalized with mPEG-NH_2_ in Tris buffer were loaded with two hydrophobic anticancer drugs, that are namely paclitaxel and curcumin, for the development of a chemotherapeutic agent [114]. A pH/ROS-responsive drug release was realized upon internalization of the modified PDA NPs by cells due to the acidic pH of cancer cells, and the increased production of ROS enhanced the drug-release efficacy due to dual response. Surprisingly, a prominent anticancer effect was observed in drug-resistant A549/TAX cell lines, indicating that, except for the improved cell internalization due to the PEG coating, the NPs could overcome the problem of multidrug resistance. Apart from conventional PEG, Zmerli et al. modified the polymer with a hydrophilic photosensitizer (PS) for synergistic PTT/PDT [115]. The idea was based on the exploitation of a ROS-responsive linker possessing a thioketal (TK) group that could be cleaved by ROS and thus release the PS. In particular, trisulfonated-tetraphenylporphyrin (TPPS_3_) was chosen as a PS for the synthesis of TPPS_3_-conjugated PEG polymer that was grafted in a second step on PDA NPs. In order to have a ROS-responsive polymer, SH-PEG_2000_-COOH was initially dimerized to protect the thiol group because it would act as a chemical anchor for the polymer to the NPs, and then the carboxylic groups were converted to acyl chloride able to react with TPPS_3_-TK-NH_2_. After the generation of the thiol group, the newly synthetized PEG could be bound to the catechol groups of PDA through Michael addition. According to the study, TK was effectively cleaved by irradiation releasing the PS and therefore leading to high PDT/PTT bimodal cancer therapy in vitro on human squamous esophageal cells. On the contrary, when a TPPS_3_-NH_2_ was used for the conjugation of the PS to PEG, a non-ROS responsive activity was observed with poor performance. Since mesoporous PDA NPs possess a higher drug-loading capacity, in some cases they are the nanoplatforms of choice for cancer therapy [116,117]. It should be noted that mesoporous PDA NPs can reach a drug loading capacity of 2.000 μg mg^−1^, which is much higher than this of non-porous PDA NPs which can span from 52 to 660 μg mg^−1^. Even if the synthetic protocol for the synthesis of mesoporous PDA NPs is different from this of solid PDA NPs, PEGylation is carried out in the same manner by mixing of the mesoporous particles with the desired PEG moiety in basic pH [118]. For example, mesoporous MNPs were synthetized using SiO_2_ NPs as a template and were post-functionalized by NH_2_-PEG in alkaline conditions. Exploiting the mesoporous structure of the NPs, they were also co-loaded with ammonia borane (AB) and doxorubicin (DOX), the most exploited chemotherapeutic drug, for successful cancer chemotherapy and acid-sensitive hydrogen-assisted ultrasonic imaging.

Apart from eumelanin mimics, the surface modification possibility of fungus-mimic synthetic analogs has also been suggested since they possess multiple quinone groups that can act as potential reactive sites for chemical modification by amine-containing molecules [119]. Only a few examples of allomelanin surface modifications can be found in the literature; however, among them, PEGylation is one of the most exploited ones for the same reasons applied to other MNPs that are increasing the stability, biocompatibility, and circulation time of NPs. It has been found that linear PEG when added to assembled allomelanin NPs, does not alter the aggregation structure, and it interacts preferentially with 4-4′ DHN dimers than with 2-2′ and 2-4′ dimers through van der Waals interaction, differently from the covalent binding seen for the PEGylation of PDA [120]. Moving to an application, PEGylated allomelanin NPs have been successfully applied for the treatment of myocardial ischemia/reperfusion (I/R) injury [121]. After the synthesis and purification of allomelanin NPs (see Section 1.1.2), they were modified with DSPE-PEG that interacted with the NPs through non-covalent bonding and thus did not alter their free-radical scavenging capacity or affect their photophysical properties. Due to the excellent and broad AOX activity of the NPs, they showed an optimal cytoprotective effect in vitro against oxidative injury while also in vivo, after being injected intravenously in mice, they were able to accumulate in damaged cardiac sites exhibiting synergistic AOX and anti-inflammatory action combined with Photoacoustic (PA) imaging. Since allomelanin NPs have been proven to possess better AOX action compared to PDA [71], offering also the possibility of visualizing through PA imaging, it is obvious that they serve a valuable alternative for the treatment of oxidative stress-related diseases.

#### 2.1.2. Metal-Functionalization

Due to the high affinity of melanin to metal ions, metal surface functionalization is a well-documented surface-modification strategy for theragnostic application since metals can act as contrast agents for bio-imaging [42]. Hence, many examples of metal-chelated MNPs are found in the literature, focusing mainly on cancer treatment and bioimaging [12,19,88]. Indeed, the incorporation of metals in PDA can be achieved in different ways: (i) deposition of PDA on metal NPs as a coating, (ii) metal post-functionalization of PDA NPs (post-doping), or (iii) synthesizing the PDA NPs in the presence of metal ions (pre-doping) [122,123]. Here we will provide an overview of the additional functionalities the different metals can endow on the pigment. As it will be shown, post-doping strategies are usually preferred for metal loading since they present an important advantage. In this case, in fact, one of the most critical parameters, that is, the size of the NPs, is controlled prior to the doping by the synthetic protocol and is not determined by the metal.

The affinity of paramagnetic metal ions to melanin can he highly advantageous for magnetic resonance imaging (MRI), and several metal-chelated melanin NPs have been tested so far as contrast agents due to their higher biocompatibility when combined with the pigment. In particular, gadolinium (Gd^3+^) chelates can enhance the signal intensity in T_1_-weighted images [124]. The metal ions can be embedded on the PDA surface simply by mixing and exhibit a relaxivity parameter (r_1_), which illustrates the dependence of the relaxation rate of the solvent on the concentration of the paramagnetic species of 2.74 mM^−1^ s^−1^, a value similar to the one of the commercially found Gadodiamide MRI imaging agent (3.47 mM^−1^ s^−1^). In a study by Chen and co-workers, different melanin-metal MRI contrast agents were investigated, focusing on the effect of the nature of metal ions on imaging properties [94]. The contrast enhancement effect of four melanin-metal chelates, Gd^3+^, Mn^2+^, Fe^3+^, and Cu^2+^, was tested in vivo and in vitro. Ultra-small MNPs were produced through the dissolution and ultrasonication of commercially available melanin in NaOH. After their purification, MNPs were PEGylated at pH = 9.5 and, after being washed to remove unreacted PEG, they were mixed with the metal ions in various ratios at neutral pH for 4 h. As expected, PEGylation was essential to prolong the circulation time and endow good water-solubility to the particles, while the affinity of PEG-MNP to the metal cations depended on their ionic radius and the ligand geometry and therefore, it was decreasing in the following manner Fe^3+^ > Cu^2+^ > Mn^2+^ > Gd^3+^. All the metal-PEG-MNPs proved to possess excellent contrast properties and high biosafety, being efficiently eliminated through renal and hepatobiliary pathways, with Gd^3+^ and Mn^2+^-doped NPs exhibiting the best results in terms of relaxivity values. Even if Gd-based contrast agents are typically used in MRI, melanin doped with this metal provides an enhanced Photoacoustic (PA) signal [125]. Since Gd^3+^ can be toxic for patients with kidney problems, in many studies, this metal was replaced by Mn^2+^ [126]. Mn-doped PDA NPs, prepared in the same way described before and being PEGylated accordingly, exhibited excellent biocompatibility, reduced toxicity, and deep-tissue penetration ability, also acting in an ideal way as contrast agents with higher r_1_ longitudinal relaxivity than well-known T_1_ MRI contrast agents [96]. Moreover, dual-modal MR/PA imaging can be realized by the same kind of NPs in combination with photothermal tumor-shrinking after intra-tumoral injection into tumor-bearing mice [127]. It should be noted that for the synthesis of L-DOPA NPs, stronger oxidants are needed, such as KMnO_4_ (see Section 1.1.2), so in this case, starting with 3,4-dihydroxy-DL-phenylalanine (DL-DOPA) as a melanin precursor, low valence states of Mn ions, reduced from KMnO_4_, are incorporated in the NPs during their formation without the need for any additional chelation processes (Figure 5) [128]. These kind of MNPs also show brilliant performance in longitudinal–transverse (T_1_–T_2_) dual-modal MR/PA imaging that can be combined with PTT for tumor ablation. Another comparative study focused on different metal ions was conducted by Zou et al. but in this case the imaging technique of interest was NIR imaging [129]. Briefly, from the six metal ions chelated on standard PDA NPs, it was found that those containing Cu(II), Fe(III), Gd(III), or Mn (III) exhibited stronger absorption in the NIR region compared to Zn(II) or Ga(III)-doped ones. This was explained by the presence of unoccupied d orbitals on the first four metal ions resulting in a lower energy band–gap for the absorption of the melanin-complex, while the d orbitals of Zn(II) and Ga(III) were fully occupied with electrons.

Due to its enhanced photothermal conversion ability upon NIR irradiation and improved T1-weighted MRI contrast, Fe(III)-chelation is a very popular alternative employed for the surface functionalization of melanin NPs as Fe^3+^ ions released in tumor sites are able to polarize macrophages towards the M1 phenotype, that is mainly responsible for a direct-host defense, killing tumor cells and activating anti-tumor immune responses [130,131,132]. Fe-doping of MNPs has been successfully applied for a variety of cancer treatments, such as bone tumor and osteolysis, combined with imaging and diagnosis; however, in most cases, the metal is co-polymerized with a melanin precursor [130,131,132]. The examples which exploit naturally extracted melanin NPs are a minority compared to those exploiting synthetic melanin, even if this approach is greener and more sustainable since usually, the sources for the pigment extraction are wastes from the food industry [50]. In an example by Liang et al., natural sepia melanin NPs, extracted from cuttlefish were purified and further functionalized with Fe^3+^ by mixing in the presence of NaCl at pH = 4 for the formation of a horseradish peroxidase-like nanozyme [133]. Since the aim was the development of a biological colorimetric sensor for the determination of the total AOX capacity (TAC) of any system, it was found that in the presence of hydrogen peroxide, the NPs were able to catalyze the oxidation of 3,3,5,5-Tetramethylbenzidine (TMB) substrate which changed color to blue. This color change was exploited for the development of the system because, upon the addition of a good AOX system, the oxidized blue substrate was reduced to colorless TMB, something that was successfully applied for the detection of the TAC in commercial vitamin tables. In another case, sepia melanin was Au-decorated by simple agitation of the natural NPs with HAuCl_4_ in water. These particular metal-functionalized NPs, except for the high photothermal conversion efficiency and good biocompatibility that are commonly seen by various MNPs, also exhibited a high X-ray attenuation coefficient for computed tomography (CT) [134].

Comparative studies are always useful for the collective understanding of the role of a metal-doping on a specific application, so regarding the sensing of genetic materials, the following metal ions, Ca^2+^, Zn^2+^, Ni^2+^, Fe^3+^, and Gd^3+^, were tested for DNA absorption on PDA for the sensing of genetic material [135]. Metals were doped on purified NaOH-prepared PDA NPs by mixing them with the PDA NPs in a buffered solution at pH 7.6, and of all the studied metals, Zn^2+^ was found to show the highest absorption efficiency of genetic material and targeted proteins. Unfortunately, doping of the NPs after their synthesis exhibited a lower performance compared to pre-doped co-polymerized metal-PDA NPs. Another kind of metal, Pt, was also utilized for gene sensing in a development by Chen and coworkers based on electrochemiluminescence (ECL). In this approach, the role of the metal-doped PDA NPs was to act as quenching-labels of ECL. In more detail, standard PDA NPs were surface-modified with H_2_PtCl_6_ and then with a streptavidin protein, which was able to selectively bind to biotin and regulate the on–off emission of specific genes [136].

Another completely different field related to the bio-application of metal-chelation on melanin that can be exploited is nano-cosmetics. Since the natural pigment of human hair is melanin, a variety of synthetic melanin-based hair dyes has been developed inspired by natural hair colors [137]. As an example, ferrous ions (Fe^2+^) in combination with PDA NPs were able to induce permanent coloration, resistant to detergents, on gray hair within 1 h [138]. A detail that deserves consideration is that iron(II) sulfate heptahydrate was dissolved in pre-synthesized PDA solutions in the presence of hair. In another example, successful hair coloration through PDA was obtained by using Cu^2+^ and H_2_O_2_ as a trigger [139]. Due to the known antimicrobial activity of Ag NPs, they were exploited in combination with melanin-inspired hair dyes in a study presented by Shen et al. (Figure 6) [58].

Various hair coloration shades were achieved by L-Tyrosine precursor-protected N- or C-terminals, something that could regulate the polymerization of the pigment and, therefore, its assembly and deposition on hair. It was shown that, by using tyrosinase and protecting the precursor, different color shade pigments could be obtained due to the incomplete enzymatic oxidation and polymerization of the melanin precursor. Combining the coloration with Ag NPs, a dual function of the hair dye was achieved in addition to sterilization and anti-inflammatory activities, except coloration, which could be obtained that may be advantageous for low immunity populations that want to treat their hair.

In closing, a possible and main disadvantage of post-doping is that usually only a low concentration of metal ions can be chelated on the MNP surface; that is why pre-doping strategies that involve the polymerization of a melanin precursor in the presence of metal ions are preferred in the many cases [123].

#### 2.1.3. Gene Functionalization

Gene–therapy (GT) is based on the delivery of therapeutic genes to patients that are able to induce genetic cell modifications in order to produce a therapeutic effect or treat diseases by repairing defective genetic material [140]. So far, GT has emerged as a promising therapeutic strategy for specific treatment of genetic anomaly-related diseases, cancer included. The success of GT relies on the specific and controlled delivery of genetic material to the target site. The highly biocompatible melanin and MNPs have been successfully employed as potential gene carriers for GT, usually in combination with PTT showing a synergistic effect [19,42,88,135,141,142]. Here we will present recent examples of gene-surface-modified melanin NPs highlighting that this surface modification approach is able to increase the target specificity of the nanocarrier significantly. First, we want to highlight that gene–surface–modification is a post-functionalization strategy in which genetic material is absorbed on the MNP surface through non-covalent interactions, usually simply by mixing because it should be effectively released at the target site. The loading of genetic material on MNP surfaces is usually monitored and verified by Z-Potential measurements, indicating a reduction in the negative surface charge upon genetic material absorption on the surface of MNPs. Importantly, MNPs have been reported to possess 100% loading capacity of nucleic acids for ratios of nucleic acid to NPs larger than one, reaching weight ratios of 40:1 MNPs/genetic material [135,143,144].

The use of short interfering RNA (siRNA) represented a real breakthrough for GT since it could silence the expression of specific genes related to a wide range of diseases. Among several examples, in a study by Feng and co-workers, PDA NPs were modified by the integration of a functional siRNA-ZIF-8 (zeolitic imidazolate framework-8) shell for synergistic GT/PTT and dual-modal imaging [145]. The surface of the PDA NPs was post-modified simply by mixing them after their preparation and purification with Zn(NO_3_)_2_·6H_2_O, 2-methylimidazole, and siRNA. The resulting nanocarrier was able to protect the siRNA from enzymatic degradation and facilitated its controlled pH-responsive release and subsequent accumulation to the targeted tumor sites. Thus, it provided a promising candidate for imagining guided cancer therapy. In a similar manner, siRNA could also be loaded on poly-l-lysine-conjugated MNPs (PLL-MNPs) for synergistic GT/PTT [143]. The developed PLL-MNPs were able to escape the endosomal barrier, and when loaded with a specific anti-survivin siRNA, an impressive inhibition effect against 4T1 tumors was observed in vivo. Continuing on the topic of siRNA-involved GT for cancer therapy, in a study by Mu et al., the most common chemotherapeutic drug, DOX, was used in combination with a siRNA specific for bone metastasis treatment [146]. These authors developed a nanoplatform based on PDA NPs, which were in the first step post-functionalized with DOX via π–π stacking and likewise with the immunosuppressant programmed death 1 (*PD-L1*) siRNA. In a second step, the siRNA-DOX-loaded PDA NPs were coated with mesenchymal stem cell (MSC) membrane, which acted as an immune camouflage for prostate cancer bone metastasis. In more detail, the membrane camouflage strategy was able to increase the stability and tumor-targeting specificity of the nanocarrier which exhibited brilliant performance for synergistic chemo-immunotherapy both in vitro and in vivo inhibiting significantly the tumor growth. Instead of standard PDA NPs, mesoporous PDA NPs have shown a better loading capacity and, therefore can be exploited in GT for an improved loading of genetic material [147]. For example, core–shell NPs were designed by Wang et al. using mesoporous PDA as a core loaded with tumor necrosis factor-α (*TNF-α*) siRNA and calcium phosphate as the shell (Figure 7) [86]. The nanoplatform was designed for the treatment of inflammatory bowel disease, and its function was based on the pH-sensitive calcium phosphate shell, which could degrade in the weak acidic environment of the lysosome and therefore release the genetic material preventing its premature liberation. Significantly, rectally administered NPs were able to protect the intestinal mucosal barrier by effective accumulation in the inflamed colon, and upon showing antioxidant performance, they could reduce intracellular ROS and inhibit the expression of pro-inflammation cytokines.

Undoubtedly, gene-editing has revolutionized biomedical research, especially with the finding of the gene-editing tool *CRISPR/Cas9*, which can precisely cut DNA, allowing the cellular DNA repair process to take place. One of the limitations in the application of *CRISPR/Cas9* is that the vectors most commonly used are lentiviruses that are poorly biosafe [148]. In a revolutionary study by Ma et al., the gene–editing tool was cloned into a plasmid-based non-viral vector loaded on PDA-based nanocarriers for enhanced target specificity [149]. In this complex approach, the vector was a Sleeping–Beauty (SB) transposon with enhanced biosafety, while for the design of the nanocarrier, PDA NPs were used as the cores, which were post-functionalized in the first step by simple stirring for one day with dexamethasone (DEX) for nuclear localization signal. After the formation of PDA/DEX NPs, they were covalently surface-modified by polyethyleneimine (PEI) for an increased gene-loading capacity and finally they were coated through electrostatic absorption with hyaluronic acid (HA) acting as a cell-targeting ligand (PDPH). The nanoplatform, showed an increased transfection efficiency and higher biocompatibility in the presence of DEX and HA coatings and by testing two different MW of PEI, 25 and 10 K, it was found that the particles coated with the higher MW polymer exhibited a higher transfection efficiency. Overall, the nanoplatform proved to be a successful non-viral vector for the delivery of *CRISPR/Cas9*, able to pass all the barriers required for genome integration that are cell and cell nuclear membranes. Moving on to another state of art development, the same kind of MNPs were surface-modified with fluorescent molecular beacons (MB) specific for intracellular microRNA-21 (miR-21) [150]. The idea was based on the use of magnetic-coated Fe_3_O_4_ PDA NPs, which could quench the fluorescence of the MB and, on the contrary, switch it on upon internalization by cancer cells due to the recognition of the tumor diagnostic market miR-21 for dual-modal fluorescent imaging and gene-silencing. All these examples indicate that GT can synergistically act with PTT, Chemotherapy, Immunotherapy, etc., while more complex surface modification strategies of MNPs can also endow imaging ability to the nanoplatforms.

#### 2.1.4. Peptide-Grafting

Due to their unique structural and functional properties, biomolecules have been widely exploited for biomedicine. Among them, peptides have not only been successfully applied for vaccination but they have also been used as drugs due to their low toxicity and high target specificity [142]. Nonetheless, the poor stability and short half-life due to enzymatic degradation of peptides is still an obstacle to exploiting their therapeutic action. In view of this, NPs have been suggested as emerging drug delivery platforms able to reduce enzymolysis, increase transmembrane absorption, and offer a controlled release of active peptides [151].

Because melanin NPs have been shown to be highly advantageous nanocarriers, they have also been successfully employed for the delivery of peptides for therapy, diagnosis [152,153], and vaccination [154]. Indeed, knowing that peptides are short chains of amino acids, connected through amide bonds, that have different side groups depending on the sequence, they can modify the surface of synthesized MNPs through binding via the side amino group improving the active targeting ability of the NPs [155]. A commonly used biocompatible peptide for this is RGD (Arg-Gly-Asp) [95], which can improve the cellular uptake and selectivity of NPs and bind to MNPs through Michael addition or a Schiff base reaction via the oxidized quinone of melanin and the amine of RGD forming a covalently grafted functional layer on MNPs [37,156]. Another possible reaction is the conjugation of the peptide amine groups with the carboxyl group of MNPs through the EDC/NHS (carbodiimide hydrochloride/*N*-hydroxysuccinimide) coupling reaction [157]. Since active targeting can significantly improve the target accumulation efficiency of a nanotheragnostic agent, Zhou et al. exploited a dual peptide (RGD and beclin 1) for target-specific autophagy-induced photothermal tumor ablation [112]. Beclin 1-derived peptide, which is able to initiate autophagy pathway also acting as a tumor suppressor, was modified with a cysteine residue and could thus be grafted onto PDA NPs through the sulfur group by mixing in an aqueous solution under an argon atmosphere. Before surface modification of the NPs with the second peptide, PEGylation took place, and then the RGD peptide (cyclo (ArgGly-Asp-d-Phe-Lys (Azide)) was reacted with a water-dispersion of PEGylated PDA NPs in the presence of ascorbic acid and CuCl_2_·2H_2_O. RGD modification was able to significantly improve the selectivity of the therapy favoring the interaction of the NPs with cancer cells overexpressing integrin α_v_β_3_ leading to their enhanced internalization and accumulation on the tumor sites. Results obtained by in vivo experiments in mice models bearing MDA-MB-231 tumors indicated that even at mild treatment temperatures (43 °C), the autophagy up-regulation promoted by PTT exhibited higher efficiency in tumor regression compared to individual treatments. From the imaging point of view, in order to overcome some limitations of many imaging-guided surgical techniques, such as PET, MRI and CT, is through the use of macroscopic preoperative views that are provided instead using intra-operative real-time imaging and the need for intravenous injection or oral uptake of the contrast agents, Liu et al. suggested an in-situ nano-spray approach for photothermal imaging-guided tumor surgery using RGD-modified PDA NPs [158]. The simplest reaction protocol in water was followed for the post-functionalization of PDA with RGD resulting in NPs with size of 280 nm and photothermal conversion efficiency of 54.27%. The novelty was based on the fact that the NPs did not have to be invasively administrated to 4T1 tumor-bearing mice, but, after just being sprayed on scissored skin of suspected tumor sites and being irradiated by a 808 nm NIR laser, they could produce a pronounced photothermal signal. Due to the peptide-conjugation, the NPs were repeatedly sprayed on the desired sites during the surgical procedure where they were able to accumulate to α_ν_β_3_ integrin overexpressed tumor cells amplifying the biological signal and converting it to thermal imaging. Unfortunately, as a drawback it should be noted that the tissue penetration depth of PDA-RGD NPs was limited. Except for cancer diagnosis, the same peptide modification of MNPs was utilized for PA imaging of another disease, rheumatoid arthritis, although a crosslinker was added for the binding of the peptide to MNPs (Figure 8) [111]. In detail, melanin NPs were PEGylated in a first step with a homobifunctional amino-PEG and then RGD was crosslinked to the PEG moiety by 4-(N-maleimidomethyl)cyclohexane-1-carboxylic acid 3-sulfo-Nhydroxysuccinimide ester sodium salt (sulfo-SMCC). The NPs were shown to be favorable contrast agents for PA imaging and, after accumulation in rheumatoid arthritis joints, advancements of arthritis treatment could be followed and aid the diagnosis. Similar to synthetic MNPs, naturally extracted melanin NPs can be post-functionalized after their extraction with the aforementioned peptide. In an example by Wang et al., melanin NPs extracted from the fungus *Auricularia auricula* were loaded with DOX and were encapsulated inside an RGD-modified platelet vesicles showing breast cancer cell-recognition properties and efficient synergistic PTT-chemotherapy eliminating tumor vasculatures for metastasis blocking [159].

Being hydrated polymeric networks, injectable hydrogels are ideal candidates for the delivery and sustained release of actives and drugs, offering the advantage of moisture maintenance required in many cases as wound healing. Regarding melanin, MNPs embedded in biocompatible hydrogels can not only act as nanocarriers of various actives, but also they can enhance the cross-linking and self-healing ability of the gels through non-covalent interactions by the catechol group [160]. In the meantime, antimicrobial peptides, loaded on MNPs, with membrane lytic activities are extremely powerful nanoplatforms as they can tackle the obstacle of antimicrobial resistance combined with the photothermal killing of microbes [161]. As an example, an antibacterial peptide ε-poly-L-lysine, able to selectively attach on bacterial surfaces, was grafted on PDA surface by simple mixing forming an antimicrobial photothermal agent for the treatment of methicillin-resistant *Staphylococcus aureus* (MRSA)-induced keratitis [162]. Combining antimicrobial peptides loaded on MNPs with hydrogels is highly advantageous not only from the delivery point of view but also for improved properties due to the hydrogel-induced self-healing [163]. A commonly used peptide loaded on MNPs is the pro-healing peptide RL-QN15 which promotes cell proliferation and migration and aids in the healing of even diabetic wounds [164,165]. In a recent case study by Sun et al., exploiting the higher loading capacity of mesoporous PDA NPs, RL-QN15 was successfully loaded on them, and then the modified PDA NPs were embedded in zinc alginate hydrogel in the design of a diabetic wound healing hydrogel [166]. According to the study, after being treated with the hydrogel, diabetic wound-bearing mice exhibited promoted healing of the damaged area through dynamic control of the inflamed microenvironment due to the regulation of macrophage activation stage and simultaneous promotion of neovascularization and collagen deposition in the problematic area. Overall, the in vivo results obtained by peptide-modified NPs indicate the potential of these agents to be applied even in humans for the treatment of chronic suffering wound conditions similar to diabetic ones.

### 2.2. Pre-Functionalization of Synthetic MNPs with Advanced Surface Properties: Co-Polymers

As made obvious in the previous chapter, even if synthetic melanin-like NPs are very useful as core materials, they are rarely used alone for bio-application since, usually, post-modification strategies take place requiring multistep synthetic protocols. In view of this, one-pot pre-polymer modification strategies have been employed as a time-saving alternative for the addition of new functionalities and the development of advantageous melanin-like materials [167]. By utilizing them, instead of simple synthetic melanin-like materials, more complex nanoplatforms can emerge with enhanced properties [168,169]. In particular, copolymers of melanin or active-enriched precursors have been presented as a novel strategy that can lead to advantageous melanin-like materials.

Fluorescence is one of the most desired properties of materials applied for bio-imaging and sensing. However, melanin is considered to be non-fluorescent with a low quantum yield. In view of this, many strategies have been utilized for the emission “switching on” of melanin-like materials through π–π stacking suppression by means of chemical-oxidation and degradation [167,170]. However, conjugation is another possible alternative that may lead to the fabrication of fluorescent MNPs. Thiol- or amine-containing moieties can be copolymerized with melanin precursors using a Schiff base or Michael addition, decreasing the polymerization degree and hindering the stacking interactions leading to an emission switch on function [171]. In light of this, DA and polyethyleneimine (PEI) co-polymer is widely exploited for the fabrication of fluorescent PDA [171,172]. As proof of principle, Zhong et al. prepared a fluorescent PDA-based probe for Cu^2+^ detection by self-polymerizing DA and covalent bonding with branched PEI (Figure 9a) [173]. The PDA-PEI copolymer NPs presented a significantly higher quantum yield than was observed for oxidized fluorescent PDA NPs. The fluorescent PEI-melanin particles, which exhibited improved photostability, were synthetized using a facile one-pot method under mild conditions in Tris buffer for 12 h at room temperature. Expanding the range of applications to disease treatment, due to the excellent metal-chelation ability of PDA-PEI shown before, the copolymer was utilized as a nano-inhibitor for Alzheimer’s treatment in a protocol demonstrated by Jin et al. (Figure 9b) [174]. In this study, it was proven that the co-polymeric PDA NPs could act as neurotoxic metal chelators able to inhibit the aggregation of *β*-Amyloid (A*β*). The inhibition ability of the nanoagent was verified against different A*β* aggregation pathways, including self-assemblies and metal-chelated aggregates, while both covalent and non-covalent interactions were demonstrated to be involved in the depolymerization of mature amyloid fibrils.

For the same purpose, instead of PEI, the essential amino acid tryptophan was used for the fabrication of fluorescent copolymeric PDA NPs [175]. In contrast to the PDA-PEI polymerization reaction carried out at an ambient temperature, the reaction between DA and tryptophan requires heating at 200 °C in Teflon hydrothermal container. Except for their proved neuroprotective effect, the amino acid-PDA NPs were proven to be excellent bioimaging agents due to their unique fluorescent properties verifying their theragnostic ability. Examining elevated temperature approaches, in another one-pot synthetic protocol, PDA/PEI NPs were reacted, forming Schiff-base structures in alkaline conditions at elevated temperature (50 °C) and tested for their capability to stabilize genetic material and their use as vesicles for nucleic acid delivery for glaucoma therapy [176]. The PDA-PEI nanocarrier was able to efficiently load a negatively charged anti-glaucoma therapeutic messenger RNA (mRNA), *miR-21-5p*, through electrostatic interactions and enhance its stability while it exhibited better biocompatibility compared to other kinds of nanocarriers as commercial liposomes. In vivo results in animal models further demonstrated the transfection efficacy of the PDA-PEI-mRNA, which significantly increased the cellular permeability of the porcine angular aqueous plexus, decreasing the intraocular pressure. Another possibility for exploiting PDA-PEI with genetic materials is to form complexes at a mass ratio of PDA-PEI NPs to DNA above five [177]. Of course, the size of the NPs greatly affects the complex formation, as PDA-PEI NPs with smaller sizes around 13 nm are able to form complexes while bigger sizes around 230 nm tend to load the genetic material on their surface something that also affects the transfection efficacy of the nanocarrier [178].

Exposure to ionizing radiation during various human activities, such as X-ray diagnosis and therapy, may have detrimental effects on human health preservation, increasing the risk of cancer occurrence. Indeed, ionizing radiation is considered a carcinogen in high doses, also affecting non-irradiated tissues [179]. Based on the fact that selenium compounds are known to be ionizing radiation protectors, the amino acid selenocysteine was used as a precursor for the synthesis of L-DOPA NPs with enhanced X-Ray protection performance [180]. Gianneschi et al. exploited a copolymerization protocol where the naturally occurring amino acid was formed in situ by selenocysteine reduction and was reacted with the oxidizing L-DOPA in the presence of KMnO_4_. The resulting selenocysteine-L-DOPA NPs exhibited superior protection against X-ray irradiation compared to both simple L-DOPA NPs and sulfur-containing pheomelanin NPs. Moreover, selenomelanin NPs were not only biocompatible but also they were able to form perinuclear caps after being incorporated by keratinocytes in a similar manner to natural melanosomes. In another study, the same group developed instead of L-DOPA, a PDA-based extraordinary radioprotector. We want to highlight that even if PDA is already a radioprotector, studies have shown that by increasing the radical content in the pigment, the attenuation of X-rays can be enhanced [181]. Based on that, a chemically functionalized melanin was synthetized by the copolymerization of DA with a stable radical [182]. In more detail, a stable nitrogen radical was synthetized by an amide-reaction of L-DOPA with 4-amino-TEMPO ((2,2,6,6-Tetramethylpiperidin-1-yl)oxyl). The radical functionality was introduced to synthetic PDA NPs by its copolymerization with DA in an ammonia solution. The introduction of the nitrogen radical significantly enhanced the X-ray protection properties of the MNPs endowing them with unique electrochemical and magnetic properties. Moreover, after being internalized by human epidermal keratinocytes, radical-enriched PDA NPs significantly scavenged X-ray-induced ROS indicating the extraordinary AOX properties of the copolymeric NPs. Based on that, in another study by Amorati et al., an inside on the advanced AOX properties of the copolymer of 4-amino-TEMPO and DA was provided [183]. TEMPO-PDA NPs showed good AOX activity against peroxyl radicals (ROO●) in water conditions under which pure PDA NPs cannot directly trap alkyl peroxyl radicals responsible for chain propagation.

Going towards other interesting pre-modification examples that lead to MNPs with new surface properties, Chen and co-workers presented PDA NPs with increased hydrophilicity synthetized by the one-pot copolymerization of DA with the alkaline natural amino acid arginine [75]. On the contrary, a copolymer between DA and polyaniline (PANI) blended with poly(vinyl alcohol), used for the construction of nanofibers, was able to reduce the hydrophilicity and increase the electrical conductivity of the nanofibers [184]. The co-polymer of DA and PANI was formed in a solvent mixture of water/dimethylformamide (DMF) in the presence of p-toluene sulfonic acid and ammonium peroxydisulfate. Since the electrospinning-formed nanofibers, prepared in a second step, were used for peripheral nerve regeneration, it was proven that the incorporation of PDA-PANI increased the biocompatibility of the nanofiber, indicating that the copolymer can act as a building block of nanofibrous neural conduit. In a similar approach, the same co-polymer, PDA-PANI, was synthetized through the same one-step chemical oxidation protocol; however, additional functionality was added. In more detail, D,L-lactaide monomer was grafted into the PDA blocks of the PDA-PANI copolymer via ring-opening polymerization resulting in a terpolymer. A nanofibrous scaffold created from the terpolymer exhibited promising physicochemical and mechanical properties for tissue engineering combined with high biocompatibility [185].

Focusing more on L-DOPA melanin precursor than DA, several impressive examples can be found in the literature related to bio-application and therapy [38]. Vaccination has always been a center of attention for humanity as a means of disease prevention, and indeed in a study by Cuzzubbo et al., the activity of L-DOPA NPs as adjuvants in peptide vaccines were investigated [153]. Upon copolymerization of L-DOPA with peptides in various ratios from 1:1 to 1:10 in alkaline and aerobic conditions, the importance of the NH_2_-terminal group was proven for the covalent binding of the peptides with the melanin precursor as when it was blocked by an acetyl residue minor binding was observed. Successfully, L-DOPA-peptide NPs were able to protect the peptides from enzymatic degradation, while in vivo experiments indicated that the nanocomplex was transferred to the lymph node for antigen presentation. Going on with important developments for human health maintenance, a co-polymer between PEG and L-DOPA was developed for bioimaging and AOX therapy of acute liver injury [186]. In the presented study, a PEGylated phenylboronic-acid-protected L-DOPA was synthetized by selectively caging the phenol position of L-DOPA with a phenylboronic acid protective group that could be cleaved by ROS, and then its amine position was linked with the same protective group through a self-immolating carbamate ester linkage and finally its carboxylic group was reacted with a PEG-NH_2_ leading to an amphiphile that could self-assemble forming NPs. The precursor was designed to be ROS-responsive, so after the exposure of phenylboronic-acid-protected L-DOPA-PEG to oxidative species, the protected groups were cleaved unprotecting the PEGylating precursor and leading to the formation of NPs in situ. This in situ formation was essential for the non-invasive monitoring of damaged tissues by photoacoustic imaging. Except for co-polymers, a pre-modified precursor can lead to the formation of NPs with stimuli-responsive bonds. As an example a cleavable L-DOPA bond was reported by the synthesis of a disulfide-linked L-DOPA dimer susceptible to glutathione (GHS) [187]. After the formation of the dimer, the precursor was polymerized in a water/ethanol mixture in the presence of NH_3,_ leading to spherical MNPs, which upon cellular uptake and GHS exposure, were disassembled through cleavage of the disulfide linkage. The NPs were surface modified with a PS, which was released upon particle disassembly, therefore, having activable photosensitizing properties for PDT, and with an epidermal growth factor receptor (EGFR)-targeting peptide for enhanced target specificity. Even if usually the co-polymerization of one melanin precursor with another functional moiety is found in the literature, an example of the copolymerization of both precursors DA and L-DOPA was reported by Niezni et al. Combining the two melanin precursors L-DOPA and DA with another monomer named 1,1,2-Trimethyl-3-(4-sulfobutyl)benz[e]indolium (In820), a triple copolymer synthesized using 1:1:1 ratio of the monomers in sodium bicarbonate buffer [89]. The developed PDA-PDO-In820 nanocarrier showed great encapsulation efficiency of various hydrophobic drugs indicating that copolymerization approaches can overcome any imagination opening new possibilities for more advanced melanin-like materials.

### 2.3. Functionalization through PDA: PDA as a Functional Coating of NPs and Surfaces

With the advent of nanotechnologies, molecular coating (MC) has become a common strategy to tune the properties of NPs and surfaces. MC is usually adopted to enhance the functionality of NPs surfaces, such as increasing the biocompatibility of nano-biomaterials or promoting site-selective targeting of nanoprobes. MC has also been exploited to donate additional functionality to surfaces, such as antimicrobial properties. In addition to the synthesis of functional MNPs, PDA has been demonstrated to fit well in these kinds of applications. In fact, its versatility in MC was reported in several studies. Most of the updated reviews focus on the application of NPs coated with PDA. Nevertheless, a systematic collection of data concerning the types of NPs that can be coated by PDA is missing. Even if a more systematic overview were given on the pre and post-functionalization of MNPs, due to the high popularity and utility of PDA MC, in this chapter, we want to present an overview of recent advances in the possibility of coating a broad range of materials with PDA including NPs and substrates [110,123,188].

For example, in recent decades, researchers tested nanomaterials to face the hardest challenges in the bio-medical field, such as cancer therapy. To fight these issues, several NPs with different chemical natures were tested, and at the same time, MC is a popular strategy for designing NPs with superior properties. In the case of organic NPs, Poinard et al. reported that PDA could be used as an alternative coating to the extensively used PEG to increase the biocompatibility and bioavailability of carbonylated polystyrene NPs for the treatment of bladder tissues [102]. In addition, Bao et al. described the methodology to coat NPs based on semiconductor polymers, such as PSBTBT, with PDA [155]. Here PDA aided the grafting of functional ligands, such as folic acid, cRGD peptide, and SH-PEG, to the PSBTBT NPs surface, and PDA guaranteed great structural stability, a higher PA resolution, and PTT amplification. Furthermore, PDA can also encapsulate small molecules, such as in the case described by Liu et al. [189]. In this study, the authors reported the possibility of synthesizing a nanoprobe by encapsulating rapamycin and chlorin e6 into a PDA shell for PTT and PDT applications, where PDA was crucial to enable PEG conjugation and PTT capability. In addition to the coated organic NPs, a wide range of articles have described the possibility of functionalizing inorganic NPs with promising applications in the medical field. For example, noble metal NPs, such as gold-based NPs, were studied in depth because they demonstrated low toxicity inside living organisms, and they are suitable for in vivo testing. Liu et al. used PDA-coated Au NPs to investigate the cellular uptake mechanism of such kinds of nanoprobes (Figure 10) [190]. The coating was performed by solubilizing DA in tris(hydroxymethyl) aminomethane aqueous buffer at pH 8.5, where Au NPs were dispersed in the solution. The thickness of the PDA layer over Au NPs was considerably affected by the reaction time and the DA concentration. The authors elucidated that the internalization process is mainly driven by dopamine receptors, such as D2DR, and the monomeric functional groups on the NP surface. PDA can function as a coating agent, even for metal oxides. Kuang et al. presented PDA-coated Gd_2_Hf_2_O_7_ NPs for multimodal imaging applications [191]. Further, PDA was demonstrated to be efficiently grafted on ceramics materials, too, as in the case reported by Zhang et al., who developed a methodology to cover mesoporous silica NPs with a layer of PDA for controlling drug-release in a pH-dependent manner [192].

PDA coating is not just related to NPs; in fact, the deposition of functional melanin-like materials on different substrates is becoming a popular way to tune the properties of surfaces. PDA coating has the big advantage of being easy to perform safely, and it can be applied to a broad range of different materials. In fact, adherent PDA films can be easily formed onto almost any surface by simply dip-coating the substrate into a solution of DA at an alkaline pH. The deposition kinetics of the PDA film can be affected not only by the nature of the buffer, such as tris(hydroxymethyl) aminomethane-based or phosphate-based ones, but also by the oxidant, such as oxygen, Cu^2+^, or ammonium persulfate [49,60,193]. Concerning its applications, the antimicrobial properties of PDA are widely known, and Zahra et al. reported the fabrication of a coating with low-fouling and nitric oxide-releasing capabilities [194]. The authors claimed a reduction in the biofilm attachment and killing efficiency for a wide range of bacteria on the functionalized surface, suggesting it as a promising therapeutic option to inhibit bacterial proliferation and biofilm formation on medical devices. Due to it owns good adhesion ability over a broad range of materials, PDA can be exploited as an adhesive to increase the adhesion between two materials. For example, Xu et al. covalently immobilized silk fibroin protein and its anionic derivative peptides onto a titanium implant surface via a PDA layer, promoting an accelerated deposition of apatite on its surface. The bioactive composite coating was observed to enhance protein adsorption and cell proliferation [195].

## 3. Perspectives

The ease of functionalization is a major advantage of melanin and its related materials, and it arises from the variety of processes that can be exploited for the functionalization. This makes melanin-like materials and NPs ideal platforms for the development of multifunctional agents for nanomedicine and, in general, for bio-applications. Multifunctionality, including a combination of therapy and diagnosis, is, in our opinion, the unique feature of nanosized agents for bio-applications. As discussed in previous sections, properly designed nano-systems can combine multiple therapeutic modalities, including PTT and PDT, and simultaneously behave as efficient contrast agents for imaging technique as MRI, PAI, or fluorescence imaging. This extraordinary versatility makes easily available very sophisticated systems, which are surely very promising for application, but also give rise to some issues that should be considered in detail and are listed below.

### 3.1. Purification and Characterization of the Produced Materials

Multi-functional melanin-like NPs can be prepared very easily, exploiting a simple one-pot pre-functionalization process, as shown in Figure 3. Nevertheless, the characterization and purification of the nanostructures resulting from the synthesis are surely complicated by the complexity of the resulting systems. Indeed, NPs behave as small solid entities which can hardly be characterized with conventional techniques such as, e.g., ^1^H-NMR or HPLC-MS. Additionally, possible modification of the composition because of the release of some functional units over time or in a specific biological environment can not be ruled out and requires a dedicated investigation. Optical techniques such as UV–Vis absorption, transient absorption, steady state, and time-resolved fluorescence spectroscopy can, in some cases, be useful for their characterization, but they may be affected by the high light scattering efficiency of melanin-like NPs and by the presence of unexpected unreacted or released molecular units.

### 3.2. Quantification and Localization of the Functional Unit

The multifunctionality of NPs is, in most cases, related to the presence of different functional, often molecular, units. The activity of these functional units is strongly related to their abundance in the NPs and their localization (e.g., onto the surface, in the core, in a specific layer). Quantifying the number of functional units and identifying their topological distribution is a fundamental issue in understanding their actual functionality. Indeed, in most of the examples of multifunctional melanin NPs reported in the scientific literature, this point is largely ignored, but it should be considered.

### 3.3. Dispersity of the Composition and Properties

Multifunctional NPs are, in general, multi-component systems assembled in a not entirely controlled way. Consequently, their synthesis does not produce a single population of NPs with a single composition but rather a family of NPs with a composition that can be more or less uniform. As a result, in the same sample of NPs, populations with very different properties (e.g., the number of a specific functional unit) can coexist. This produces a polydispersity of the properties that affect the overall behavior of the NP sample. In simple words, a sample of NPs may contain a fraction of NPs suitable for a specific application and other populations totally un-efficient or even with undesired properties. Separation of the different populations of NPs should be, even though difficult, very important and deserves to be considered.

### 3.4. Comparison of the Performances

Although, thanks to functionalization, several kinds of NPs have been synthesized and found important bio-applications, the actual quantification of their activity and multifunctionality is very difficult. In addition to the existence of a multiplicity of melanin-based nano-systems for the same application (e.g., PTT in combination with PAI), an actual comparison of the performance of different platforms is rarely considered. In our opinion, some benchmark experiments and parameters should be identified to quantify the efficiency of the NPs and also distinguish the different functions.

### 3.5. Effect of the Functionalization Strategy

Different functionalization strategies are expected to produce different structures and properties. For example, upon pre-functionalization, a consistent fraction of the functional units is expected to be incorporated in the melanin-like structure rather than on the surface. Depending on the functional unit, this may result in advantageous protection from the external environment (e.g., for a drug) or its undesired deactivation (e.g., for an aptamer). Functionalization strategy is hence fundamental, and when possible, different strategies should be compared.

### 3.6. Effect of the Functionalization Order

While pre-functionalization is usually a single-step process, post-functionalization may occur in multiple steps. Changing the order of these steps can lead to NPs with different compositions and properties. This effect should be taken into consideration.

### 3.7. Cross-Talk between the Functions

Considering multi-functionality, cross-talk between the different functions is, in our opinion, a major issue. Indeed, a functional unit that performs efficiently for a specific function alone does not necessarily maintain this behavior in a potentially multi-functional system. This concept can be explained clearer with an example: melanin is a unique agent for PTT and PAI, but these features can also be implemented in NPs able to perform PDT by functionalization with a proper molecular PS. On the other hand, melanin, due to its optical and electronic properties, can quench the PS electronic excited state making it totally ineffective for ROS generation. This example demonstrates that functions are not simply additive in NPs and that, in general, PTT activity is expected to hamper PDT efficiency in multi-functional systems. Generally, different functions can interfere with each other. This interference or cross-talk can be, in part, attenuated by a proper design but should always be considered and possibly quantified.

### 3.8. Possible Use of Different Melanin-like Materials

It is widely ignored in the scientific community that multi-functionality may also be achieved by using different kinds of melanin-like materials. Most examples of multifunctional nano-systems are based on PDA, while other species, such as allomelanin, present, at least in part, different properties. In general, melanin-like materials different from PDA deserve more consideration in the field of bio-applications.

### 3.9. Cost Evaluation

Undoubtably, synthetic melanin-like materials provide the possibility to choose the desired size by tunning the synthetic protocol; thus, a higher control and narrow size distribution can be obtained. For the synthesis of PDA, Dopamine·HCl is utilized as a precursor, which is a commercially available and relatively low-cost reagent (3.2 USD per gram from Sigma-Aldrich), and thus one liter of DA solution (1 mg/mL) can be produced quite economically [84]. Regarding natural sources, as mentioned in Section 1.1.1, natural melanin can be extracted from various sources, including cuttlefish ink, black sheep wool, human or animal hair, bird feathers, black oat, black garlic, yeasts, fungi, and mushrooms. The extraction procedures usually involve solvents commonly found in the laboratory, as described in a comprehensive review by Pralea et al., therefore, in this case, the cost is strictly related to the melanin-source of choice [196]. As an example, sepia ink is also utilized by the food industry and is a much more cost-efficient source for the extraction of melanin, requiring only water purification, compared to melanin extracted from human or animal hair where harsh alkaline extraction conditions are required. However, commercial melanin formulations of either natural or synthetic origin are still expensive, and indeed the cost-efficient production of melanin at a large scale presents a big challenge for the scientific community. For this reason, microbial melanin has emerged as the most low-cost, green, and sustainable approach based on a circular economy for the large-scale production of melanin [197]. This approach is based on the exploitation of agro-industrial wastes and byproducts for the production of nutrient-rich fermentation media for microbial melanin synthesis. Even if the market demand for melanin continuously increases, up to now, there is not standardized extraction method for the natural pigment, while commercialization is still limited.

## 4. Conclusions

Due to the presence of various chemically active groups on the surface of both natural and synthetic melanin, numerous classes of functional units can be bound on the surface of the pigment, including peptides, genetic material, polymers, and metals to produce multifunctional nanoplatforms with enhanced specificity, target selectivity, imaging possibility, and synergistic therapeutic outcomes. A particularly attractive feature is the ease of modification that melanin confers since, in many cases, functional moieties can be bound on the surface of the pigment simply by mixing. Due to this high versatility combined with ease of preparation, high biocompatibility, and unique optical and electronic properties, artificial melanin-like materials have been successfully applied to a wide range of bio-applications, including therapy and cosmetics. At the same time, melanin extracted from natural sources, especially from cuttlefish, possesses inherent biocompatibility and has shown promising results in cancer therapeutics. Here we gave an overview of all the pre- and post-functionalization possibilities of natural and synthetic NPs summarized in Table 1, also highlighting the versatility of PDA as a ‘primer’ coating agent for further functionalization of almost any nanomaterial or substrate. In particular, we presented and discussed some of the most recent and important examples focusing on the multiplicity of therapeutic and diagnostic functions achievable in melanin-like NPs by the appropriate functionalization. Finally, we identified the most critical issues in the design of melanin-based multifunctional nano-platforms listing the critical points that the scientific community should pay more attention to.

Considering the tremendous progress in the use of MNPs achieved by the scientific community for the design of multifunctional nano-systems, the overview of the possibilities that can emerge through synthetic and natural melanin modification is valuable for accelerating research and aiding the advancement of melanin-based materials.

## Figures and Tables

**Figure 1 ijms-24-09689-f001:**
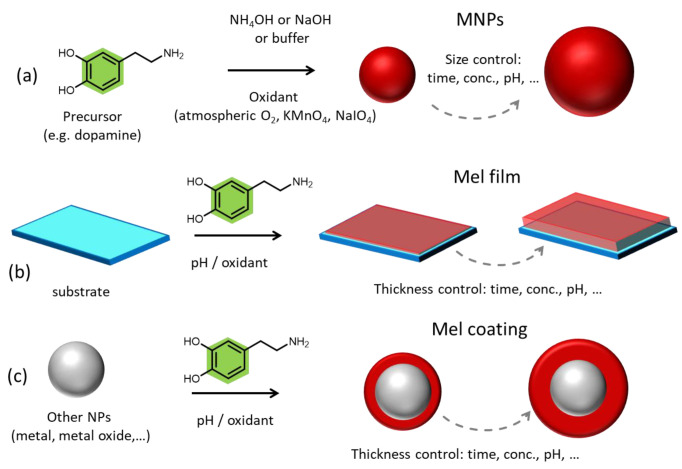
Scheme of the synthesis of (**a**) artificial melanin-like NPs (MNPs); (**b**) melanin-like films; (**c**) melanin-coated NPs.

**Figure 2 ijms-24-09689-f002:**
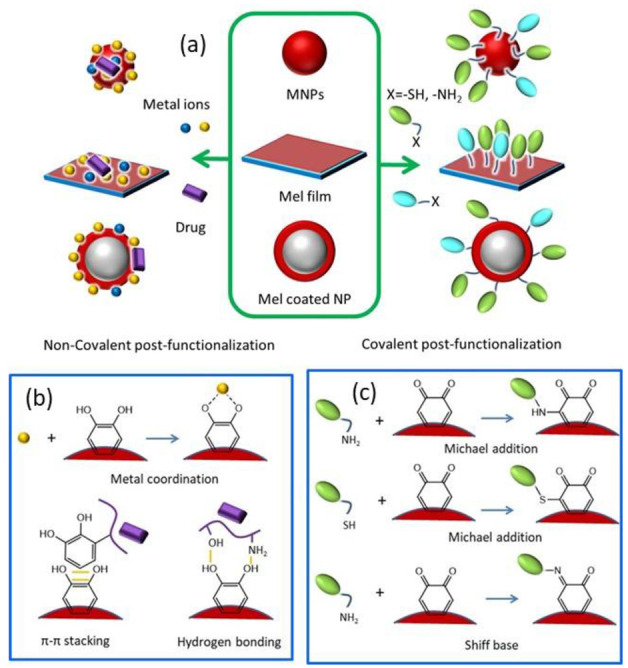
(**a**) Methods of covalent and non-covalent post-functionalization of melanin-like NPs, melanin-like films, and melanin-coated NPs. (**b**) Examples of non-covalent interactions. (**c**) Example of reactions for covalent functionalization.

**Figure 3 ijms-24-09689-f003:**
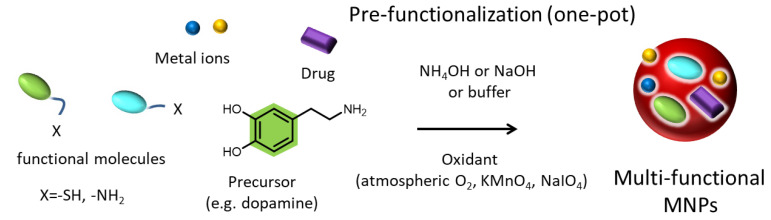
Scheme of the one-pot process for the production of multi-functional melanin-based NP through pre-functionalization.

**Figure 4 ijms-24-09689-f004:**
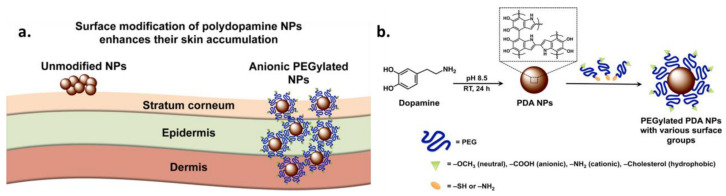
(**a**) Schematic illustration of the skin accumulation capacity of unmodified and anionic PEGylated PDA NPs, (**b**) Overview of PDA NP synthesis and surface modification with various PEGylated ligands to produce neutral (PDA@PEG), anionic (PDA@PEG–COOH), cationic (PDA@PEG–NH_2_), and hydrophobic (PDA@PEG–Chol) NPs [108].

**Figure 5 ijms-24-09689-f005:**
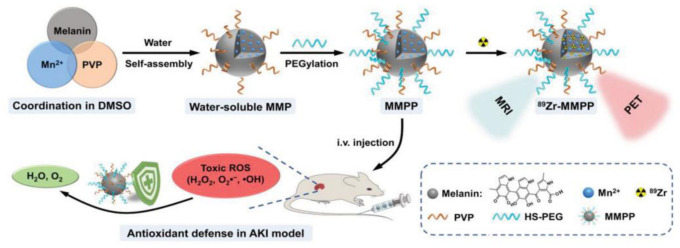
Schematic illustration of Mn-chelated MNPs illustrating their activity as a naturally antioxidative platform for PET/MR bimodal imaging-guided acute kidney injury therapy [128].

**Figure 6 ijms-24-09689-f006:**
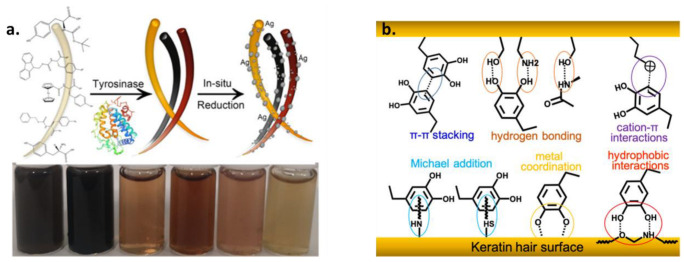
Schematic illustration of (**a**) synthetic procedure for the typical Ag NPs deposition of tyrosinase-colored hair, and (**b**) the possible mechanism of melanin-like pigment deposition to keratin hair surfaces [58].

**Figure 7 ijms-24-09689-f007:**
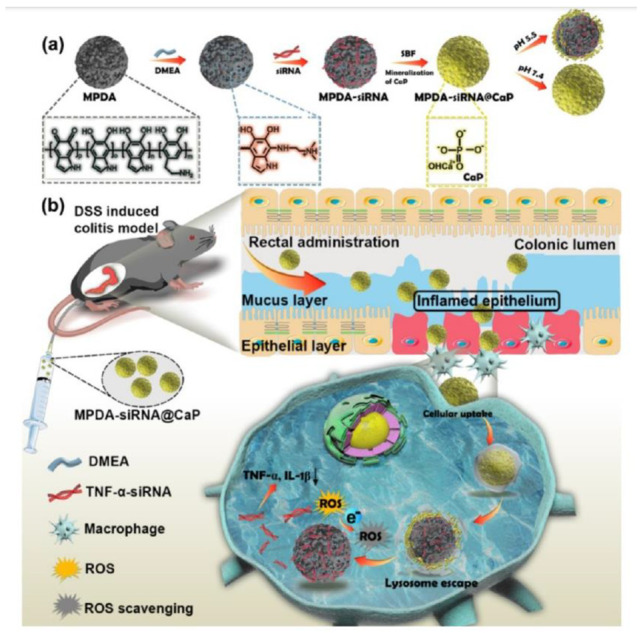
Schematic illustration of (**a**) the preparation of mesoporous PDA NPs loaded with siRNA and capped with a pH−sensitive calcium phosphate shell, (**b**) the in vivo application of the NPs for ROS scavenging synergistic gene therapy of inflammatory bowel disease [86].

**Figure 8 ijms-24-09689-f008:**
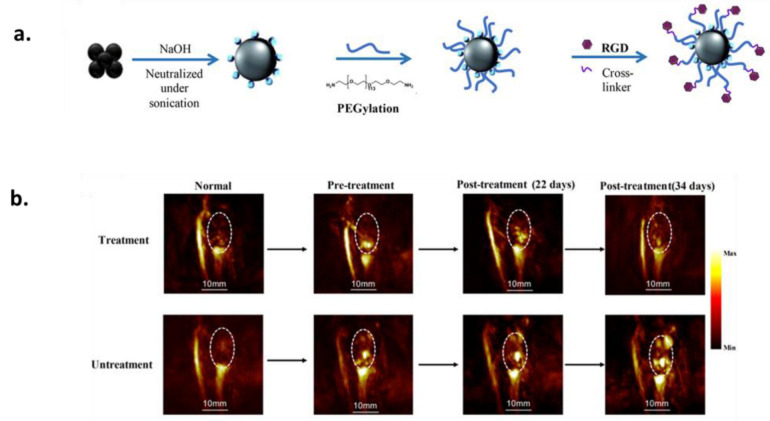
(**a**) Schematic illustration of the synthetic procedure of NaOH-generated MNPs that are PEGylated and loaded with RGD peptide, (**b**) PA images of treatment and un-treatment group after 3 h after intravenous injection of PEGylated MNPs functionalized with RGD (scale bar is 10 mm) [111].

**Figure 9 ijms-24-09689-f009:**
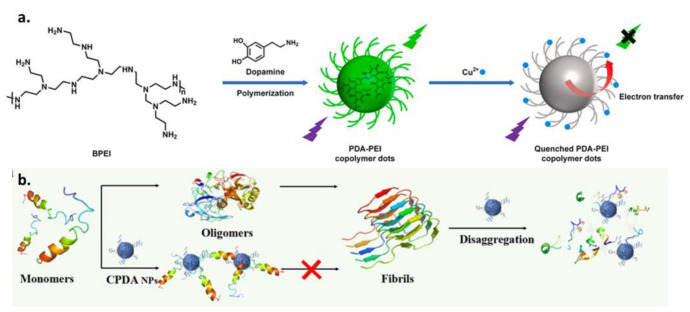
Schematic illustration (**a**) of the synthesis of PDA-PEI copolymer dots and the Cu^2+^ determination based on the PDA-PEI copolymer dots, (**b**) of the action of PDA-PEI in the modulation of *Aβ* Self-Assembly [173,174].

**Figure 10 ijms-24-09689-f010:**
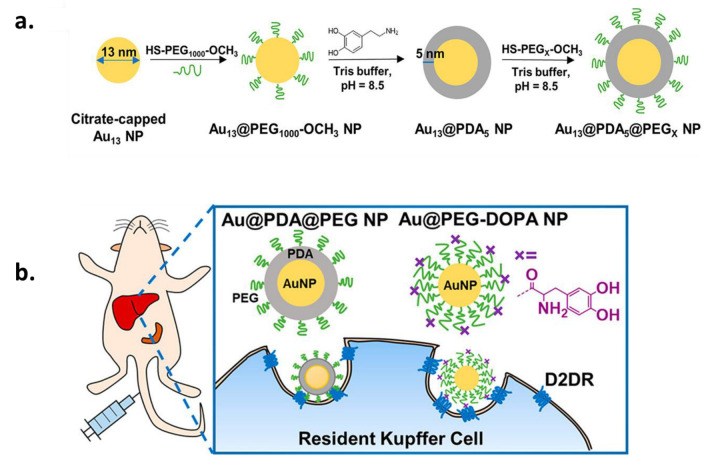
Schematic illustration (**a**) of the binding of PDA-coated Au NPs with dopamine receptor, (**b**) synthesis of Au NPs and coating with PDA [190].

**Table 1 ijms-24-09689-t001:** Summary of the main Post-Functionalization and Pre-Functionalization Strategies of Melanin NPs, including the relevant studies discussed in this review.

Post-Functionality		References
PEG	mPEG	[120,121]
	PEG-NH_2_	[102,114,118]
	PEG-SH	[108,110,112,126,155]
	NH_2_-PEG-NH_2_	[94,96,108,111]
	HS-PEG-COOH	[97,108,115]
	PEG-Chol	[108]
	FA-PEG-NH_2_	[109]
Metal	Cd^3+^	[94,124,125,129,135]
	Mn^2+^	[94,96,126,127,128]
	Mn^3+^	[129]
	Fe^3+^	[94,129,133,134,135,138]
	Fe^2+^	[138]
	Cu^2+^	[109,129,138,139]
	Zn^2+^	[129,135]
	Ga^3+^	[129]
	Ca^2+^	[135]
	Ni^2+^	[135]
	AgNPs	[58]
	Pt^4+^	[136]
Gene	siRNA	[86,110,143,145,146]
	*CRISPR/Cas9*	[149]
	miRNA	[150]
Peptide	RGD	[37,111,112,156,157,158,159]
	beclin 1	[112]
	ε-poly-L-lysine	[143,162]
	RL-QN15	[164,165,166]
**Pre-Functionalization**		**References**
**Precursor**	**Functionality**	
DA *	PEI	[171,172,173,174,176]
	Tryptophan	[175]
	4-amino-TEMPO	[181,183]
	Arginine	[75]
	PANI	[184,185]
L-DOPA	Selenocysteine	[180]
	PEG-NH_2_	[186]
	pOVA30 **,gp100 ***, Acetyl-R VIYRYYGL	[153]
	DA and 1,1,2-Trimethyl-3-(4 sulfobutyl) benz[e]indolium	[89]

* DA = Dopamine, ** pOVA30 = SMLVLLPKKVSGLKQLESIINFEKLTKWTS, *** gp100 = KVPRNQDWL.

## Data Availability

Data sharing no applicable.

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
