# Peer review of "Functionalization of and through Melanin: Strategies and Bio-Applications"

_ijms, 2023, doi:10.3390/ijms24119689_

Round 1
Reviewer 1 Report
The work by Alexandra Mavridi Printezi et al. focused on an exhaustive review of melanin-based materials, describing the different functionalization mechanisms and the main application in bio-related field. The topic itself is of special relevance being an emergent research focus. The sections and the discussion of the reported examples have been carried out critically, comparing the advantages and disadvantages of the different systems and the approaches used. In addition, the review incorporates a perspectives section which gives it greater quality. Overall, the review is well written and organized, being of great interest for its publication in the International Journal of Molecular Sciences.
I suggest minor issues:
1) Include a table with the reported examples classified by the main features highlighted in the manuscript.
2) Grammar and English check throughout the manuscript. Some of them:
“Apart from eumelanin mimics, the surface modification possibility of fungus-mimic synthetic analogues has have been also suggested…”
“Moreover, dual-modal MR/PA imaging can be realized by the same kind…”
“Theses was were explained by the presence of unoccupied d…”
“H2PtCl6 and then with a streptavidin protein ,…”
“…through PDA was obtained by using Cu2+ and H2O2 as a trigger [136].”
“…could be obtained that may be advantageous for low immunity populations that want to treat their hair.”
“…MNPs was utilized for PA imaging of another diseases, rheumatoid…”
“…strategies take place requiring multistep synthetic protoclos.”
Author Response
Please see the attachment for our response to the reviewer's comments.

Reviewer 2 Report
Dear Authors,
I have read the manuscript „Functionalization of and through melanin: strategies and bio-applications”. I consider that it is an interesting paper, well structured, with a lot of useful information for the readers. The presented figures are suggestive for the field addressed in the paper.
The ease of functionalization is a major advantage of melanin and its related materials, and it arises from the variety of processes which can be exploited for the functionalization. The perspectives of using these types of materials in nanomedicine and beyond, are promising.
In conclusion, I agree with the publication of this article in the International Journal of Molecular Sciences.
Some corrections:
1.– I noticed that, in References section, the name of the journal is missing for the references 7, 12, 13, 31, 41, 55, 68, 78, 79, 83, 89, 90, 96, 119, 122, 125, 148, 164.
2. Fig.8a – as a recommendation, if possible, the formulas from the second row to be more visible.
Author Response

(The authors gave the same response as above.)

Reviewer 3 Report
Overall Comments
The Review Manuscript by Mavridi-Printezi and colleagues focuses on melanin-related nanomaterials and their impact on nanomedicine and bio-imaging. Melanin is a robust and versatile platform for nanoparticle surface functionalization and is of interest due to ease of functionalization and wide ranging biomedical applications. The manuscript is comprehensive and well written. Once the authors address a few minor points (below), the manuscript is suitable for publication.
Specific Comments
1.) What are the costs associated with melanin extraction from natural sources vs. synthetic production? The review would benefit from a brief discussion of costs related to melanin NP production. This could be added to the Perspectives section.
2.) How uniform are the melanin-NPs and what degree of uniformity is achieved with surface functionalization?
3.) Lines 149-150, what is meant by high drug loading? Could the authors please clarify how many single drug molecules can be loaded/functionalized onto melanin NPs?
4.) Small point: Line 302, please change “easy” to “ease”
5.) 2.1.1: What are typical circulation half-lives of unmodified vs. PEGylated MNPs?
6.) Small point: Line 654: please change patience to “patients”
7.) How many genes may be incorporated into a single MNP?

Author Response

(The authors gave the same response as above.)
